# Behind the Frontlines: Insights for Supporting Mental Health and Staff Retention in the Long-Term Care Workforce

**DOI:** 10.3390/healthcare13010040

**Published:** 2024-12-29

**Authors:** Chelsea B. Smith, Karen Lok Yi Wong, Sheila Dunn, Mario Gregorio, Lily Wong, Polly Huynh, Lillian Hung

**Affiliations:** 1Innovation in Dementia and Aging (IDEA) Lab, University of British Columbia, Vancouver, BC V6T 1Z4, Canada; 2Community Engagement Advisory Network, Vancouver Coastal Health, Vancouver, BC V5Z 4H5, Canada; 3Richmond Home and Community, Vancouver Coastal Health, Vancouver, BC V5Z 4H5, Canada

**Keywords:** long-term care, workforce, staff, nursing, mental health, COVID-19, qualitative study, thematic analysis

## Abstract

**Background/Objectives**: Canada’s long-term care (LTC) sector is struggling with a significant staffing crisis related to shortages, high-turnover rate, and challenging working conditions. The COVID-19 pandemic exacerbated these issues and emphasized the need for improved mental health support for LTC staff. Understanding and addressing the wellbeing of staff is important for ensuring quality of care and promoting a positive work environment for a healthy workforce. This study explored staff experiences in Canadian LTC homes during the COVID-19 pandemic and offers staff-driven recommendations to support staff mental health and retention moving forward. **Methods**: We applied the Collaborative Action Research (CAR) methodology to explore practical strategies with LTC staff to inform actions for change. Sixteen staff members working in two large urban Canadian LTC homes were interviewed using remote videoconferencing and phone calls to conduct one-on-one interviews. Thematic analysis was performed. **Results**: Our analysis identified four themes: depletion, lack of support, providing resources and sense of community. The SUPPORT framework was created based on staff recommendations to improve LTC staff mental health and retention. **Conclusions**: Urgent attention is needed to support the LTC workforce through practice change and improved policy. The implementation of comprehensive frameworks such as SUPPORT can play a pivotal role in fostering staff resilience, enhancing job satisfaction, and promoting a healthy workforce for aged care.

## 1. Introduction

Working in LTC is uniquely challenging due to numerous factors including high physical and emotional demands, end-of-life care, understaffing, high turnover rates, and limited resources. These factors affect the wellbeing of staff, resulting in poor worker retention and a strained LTC workforce. These challenges were exacerbated by the COVID-19 pandemic and emphasized the need for better mental health supports for staff. Fostering a healthy LTC workforce is critical in enhancing staff retention and promoting high-quality, compassionate care for residents. A resilient LTC workforce is necessary to meet the increasing demands of the aging population.

The impact of the COVID-19 pandemic on work in LTC includes increased workloads and stress, staffing shortages [1], increased infection control [2], emotional and mental strain of workers, visitor restrictions [1], challenges meeting needs of family/friend caregivers [3], frequently changing policies and procedures [4], and coping with grief over loss of residents [5]. In 2020, more than 86% of Canadian LTC facilities reported staffing related issues with 71–77% increase of overtime hours and absent workers [6]. Restriction of family visitors who often help provide care and social connection for residents results in increased workloads for LTC staff and an unmet social need from residents that staff were left to try to meet [7]. The pandemic also created challenges and conflict between some staff and family/friend caregivers, such as managing conflicting demands from public health policy and caregiver wishes [3]. Some of the changes in policy during the pandemic in parts of Canada included restricting workers to one LTC facility to reduce spread of the virus [8]. There was a period where workers were provided additional ‘pandemic pay’ given increased demands and sacrifice of the job, although this was not a permanent change.

Understanding staff experiences during the COVID-19 pandemic reveals key information on the gaps in the LTC system during a particularly challenging period that can inform and improve the system moving forward. There is growing literature that explores the experiences of healthcare workers in LTC during COVID-19. Among this literature, there is a scoping review describing the experience of LTC staff across 12 countries [1]. The review highlights common themes of moral distress, feelings of pressure, and emotional exhaustion to bring emphasis to the need of improving LTC workers mental health.

Studies from different parts of the world concerning the mental health of LTC workers were conducted during the COVID-19 pandemic. Using a self-reported questionnaire with validated mental health scales and semi-structured interviews, the study by Tebbeb et al. in 12 nursing homes in France found a sense of powerlessness, a feeling of limited time and staffing, anxiety, depression, and post-traumatic stress among staff during the pandemic [9]. Takahashi et al. conducted a study about staff mental health in 284 nursing homes in Japan using a survey during the pandemic [10]. They found symptoms of anxiety and depression among staff, especially if they were working with family caregivers who were facing mental health issues. Bahr et al. used a questionnaire to ask administrators in residential care facilities in California in the United States about the experience of their facility during COVID-19 [11]. Key factors contributing to stress levels among staff included physical, emotional, and mental challenges of the pandemic, staffing issues, and changing public health guidelines. Cockshott et al. interviewed care home staff in the United Kingdom regarding their emotional challenges during COVID-19 [12]. They found staff felt anxious and distressed due to the uncertainty of the pandemic, the illness and death of residents, worries about their own health, and their hard work not being recognized. They also felt stress because of the increased care load and reduction of external support. Fisher et al. interviewed care home staff in New York City, United States, about factors related to their well-being during COVID-19 [13]. They found that the stress came from the changes of the virus, workplace, home, and work-life balance. Staff also recommended how the care home leadership could improve their well-being, such as being more empathetic to their challenges, better staffing, transparent communication, and adequate care supplies. In a cross-sectional study in Taiwan about stress and resilience among nursing assistants during the pandemic in 21 long-term care institutions, Kuo et al. found those working three eight-hour shifts reported less stress than those working two 12-h shifts [14].

The importance of resilience among LTC staff during the pandemic has been discussed in numerous studies. A negative correlation between stress and resilience has been reported during the pandemic [14]. In one study in Canada, resilience was described as “the ability to adapt to changing circumstances and protocols, while also maintaining a positive attitude and uplifting spirits during times of adversity” [15]. Resilience is subjective and complex with some part of resilience that can be attributed to individual attributes and character. However, there are external factors and support that can build up or wear down resilience over time. In the study by Danish et al., the factors that LTC staff identified as affecting their resilience included available resources, leadership and management, social supports, resident morale, and personal attributes [15]. In another study on resilience, LTC nurses were fueled by their professional identity as a nurse [16]. In this study by Connelly et al., thematic analysis suggested themes could either drain or fuel resilience among nurses, including the dynamic role as a nurse, preservation of self, banding together and a sense of support from leadership. For example, for preserving self, some staff had views focused on compartmentalization in response to this element of their work, while others viewed their preservation of self was in the form of quitting. One study identified ways that LTC nursing staff build resilience, which is characterized as perception or strategy [17]. Perceptions included efforts to understand the situation, taking responsibility of the resident, reflecting on the value of life, and thinking about one’s strength and areas of improvement. Strategies included evaluation of self and environment, self-care, finding concrete solutions to problems and self-growth.

While COVID-19 has had a profound global impact on the LTC workforce, it is important to examine each country individually, considering the unique interplay of local and national healthcare policies, outbreak response, and resource availability. Statistics Canada released a recent report emphasizing the need to investigate experiences and outcomes of the pandemic related to LTC staff health, well-being, and working conditions and offer important lessons moving forward [18].

Available studies in Canada that focus on staff wellbeing during the COVID-19 pandemic have collected data from surveys and interviews. The prevalence of anxiety and depression was shown to increase among nurses in British Columbia related to the onset of the pandemic [19]. Statistics Canada report on a 2021 survey on healthcare workers experience during the pandemic [18]. Data was collected from 2051 LTC staff. Immigrant health workers were more likely to experience precarious work (e.g., incomes loss, unpaid leave) and precarious work in general was associated with stress and poor health. Reynolds et al. administered an online survey among 70 LTC staff and managers in central Canada [4]. They found moderate levels of stress particularly related to rapid changing guidelines, increased workload, struggling to meet the needs, fear of the virus, and concern over a negative view of LTC.

Few studies have interviewed LTC staff directly. Savage et al. interviewed 21 Canadian nursing home leaders during the early waves of the pandemic [20]. The authors report four themes including responsibility to protect, overwhelming workloads, mental and emotional toll, and moving forward. They emphasize the mental distress LTC leaders faced and administrative chaos related to the pandemic. Connelly et al. interviewed 40 registered practical nurses working in LTC homes in Ontario and report on a bimodal process representing differing levels of resilience among the nurses, where some were able to maintain resilience while others were not, and how this resilience was drawn from themselves as individuals [16]. They reported a need for supports for self-care, work-life balance, promotion of connection amongst staff, team-based care, and collegial support in problem solving. Titley et al. interviewed care aides in LTC in Alberta and British Columbia about their experiences during the pandemic [21]. The authors found care aids experienced mental and emotional distress from different factors, including imposing resident isolation, grief about the loss of residents, worry about the spread of COVID-19 and becoming infected, and a lack of staff.

There has been a particular interest in examining moral distress and injury related to the pandemic. Haslam-Larmer et al. administered a survey to 227 staff members working in LTC in Ontario to examine changes in moral distress during the pandemic [22]. More than 80% of staff working with people with dementia reported an increase in moral distress since the start of the pandemic. Factors contributing to moral distress included lack of activities and family visits, understaffing, high staff turnover, and harmful policies and procedures. Over 50% of participants reported physical exhaustion, anxiety, powerlessness, and guilt related to moral distress. Reynolds et al. interviewed LTC workers in Ottawa and Manitoba about moral injury and administered clinical diagnostic assessments [23]. They report on the experiences and impact of moral injury in LTC during the pandemic and highlight the need for mental health supports. They also demonstrate anxiety and eating disorders were among the more common psychiatric disorders classified.

Exploring a variety of Canadian LTC staff perspectives from different care homes, healthcare authorities, and provinces helps us better understand experiences as we work towards improving staff retention, mental health, and quality of care. This paper presents one-on-one interviews with a diverse range of staff members such as nurses, recreation staff, music therapists and management from two different LTC homes in British Columbia. We compare findings to existing literature and highlight new insights. This work adds value to the literature because in addition to characterizing the experiences of LTC staff, it focuses on staff-driven recommendations to make improvements in policy and practice. We specifically asked the staff what they need to feel more supported while working in LTC and collected detailed responses to inform a framework for change. This work adds value to strengthen the literature and encourage a resilient LTC workforce, while offering tools to do so.

## 2. Materials and Methods

### 2.1. Study Design

The study presented is part of a larger study investigating loneliness among LTC residents [24]. As part of the study, staff members were interviewed to better understand resident wellbeing during the COVID-19 pandemic. In early stages of the study, it became clear to the research team that LTC staff faced substantial challenges during the pandemic and their mental health suffered. Furthermore, initial data suggested the wellbeing of residents strongly relates to the wellbeing of staff working in the LTC home. In response to these findings, the team began to investigate staff experiences specifically.

This study utilized the Collaborative Action Research (CAR) methodology [25], a cyclical process that includes planning, acting, observing, and reflecting phases. After study approval and participant recruitment, key issues and priorities were identified in early cycles to help guide subsequent steps. The CAR methodology focuses on engaging stakeholders to enhance appreciation of complex issues and explore solutions. This involved working with team members from different backgrounds, including a social worker, an older adult living with mild dementia and family members of LTC residents. One of the principal investigators of the study is part of the older adult mental health team in the health authority and brought shared understanding to the research team discussions. This paper presents data and analyses of experiences of LTC staff including identified issues and suggested solutions to improve staff mental health and retention. The insights gained will be utilized during the next phase of CAR to deploy actions and change policy in collaboration with stakeholders.

### 2.2. Research Setting and Participants

This research was conducted in two urban LTC homes in British Columbia, Canada. Both facilities had a large COVID-19 outbreak (i.e., more than 26 cases of COVID-19) and resident deaths between 2020 and 2021 [26]. These facilities were chosen since they had large outbreaks and represented extreme cases to show the severe impact of the COVID-19 pandemic. The residents living in these LTC homes also had complex needs and required 24-h nursing care. These facilities require a strong workforce and further demonstrate the impact of the pandemic on staff.

Staff members from different roles were invited to participate in the study, including frontline workers, administrators, and managers. The inclusion criterion was English-speaking staff who were able to comprehend the study’s purpose and procedures; there were no exclusion criterion. Convenience sampling was used for recruitment. The recreation staff helped display study recruitment posters and informed staff members about the study. We reached data sufficiency or “information power” [27] to answer the research questions after interviewing sixteen staff members. Two staff members withdrew from the study due to busy schedules.

### 2.3. Research Team

The research comprised seven females and one male (MG). The team included traditionally trained academics, an older adult living in the community with mild dementia (MG) and two family partners (i.e., have relatives living in LTC; SD and LW). The team members MG, SD, and LW encouraged a patient-oriented approach to data collection and analysis. Patient-oriented research emphasizes engaging patients, caregivers, and families as research partners [28]. These team members contributed to data interpretation and were part of team discussions. They offered key insights into the complexities of LTC. Interviews were led by team members trained and experienced in qualitative research including CS (BSc, Research Assistant), KW (MA, MSW, Research Assistant), and LH (RN, PhD, Assistant Professor).

### 2.4. Data Collection Procedure

Semi-structured one-on-one phone interviews were conducted with LTC staff between September and December 2021. Guiding questions are provided in Appendix A. Interviews were arranged based on staff schedules and availability. Interviews were completed in private offices or another quiet and private location. Interviews were 30–60 min long and were audio recorded and transcribed and anonymized. Field notes were also taken to capture additional observations.

### 2.5. Data Analysis

Interview transcriptions were read by team members and discussed during team meetings. Ideas and interpretations from all team members were encouraged. This allowed insight from individuals with different backgrounds (i.e., clinical researcher, older adult living with dementia, family partners, social worker, research assistant) to contribute to the analysis. Thematic analysis was employed using an inductive coding approach [29]. CS generated initial codes and collated relevant data for each code. The codes were then grouped into categories, which were further grouped into descriptive themes. Theme names included a staff quotation that was representative of the theme codes, categories and themes in the study are described further in Appendix B. Themes were discussed as a research team to gain analytic consensus. A grounded approach was used throughout the analysis.

### 2.6. Ethical Considerations

The study received approval from the University of British Columbia’s Research Ethics Board and the regional health authority. Prior to conducting interviews, participants provided oral and written consent. They were informed of their right to withdraw from the study at any point.

## 3. Results

Sixteen staff members were interviewed (nurses, care workers, music therapists, recreation staff, administrators, and directors). Table 1 summarizes the descriptive characteristics of the participants.

Thematic analysis revealed four themes. The first two themes characterize what contributed to worsening staff mental health and retention, including (1) depletion (“staff are drained”) and (2) lack of support (“make sure your staff feel supported”). The final two themes encompass the tools that helped improve staff mental health and include (3) providing resources (“when you had the extra staff then you were able to finish and spend more time with everybody”) and (4) sense of community (“It’s those people that they see every day. It’s your work family”).

Theme 1. Depletion

Staff felt mentally and emotionally depleted during the pandemic due to compounding factors that exacerbated existing challenges in LTC. The rapid spread of COVID-19 led to staff members being sent home due to exposure, straining the already understaffed workforce. One staff member recalled, “Care aids were sent home, not because they had the virus, but because they were exposed to it. There was just one care aid for about 27 residents”. Additionally, some staff quit or took a leave of absence due virus fears and/or exhaustion. Increased safety measures and care needs from sick residents further increased workloads. During visitation restrictions, staff lost help from family members who normally assist residents. Help from visitors was important prior to the pandemic because staff shortages have been a long-standing problem. One staff member stated, “What happened with the pandemic is actually an eye-opener as many of the issues have already been going on. I think it just magnified it”. Together, these factors resulted in staff taking on multiple roles, working long shifts (e.g., over 24 h), and feeling a sense of hopelessness.

Staff emphasized the importance of their relationships with residents and how these connections are why they entered the field. Unfortunately, the increased workload and fear of virus transmission forced staff to be more task-oriented and reduce social interaction with residents. Staff also felt a responsibility to residents and wanted to prevent bringing the virus into the LTC facility. Staff members would limit their “bubble”, i.e., the people they saw during the pandemic, to residents and staff within the facility. One staff member described “I tended to isolate myself from any personal bubble”. Many staff did not see other people, including their families, outside of work.

Staff also struggled with feelings of fear and guilt. Staff feared infection of the virus, including visits from unvaccinated visitors. They were afraid of spreading the virus to their families at home, especially those who were vulnerable. Some staff members also struggled with guilt because they felt they were letting residents down (e.g., “there’s nothing I can do to help them”). They expressed how passionate they were about their jobs and how much they wanted to help residents, who were clearly suffering (“they were really suffering. I could hear mourning, I could hear them cry, asking for help”). Staff members had to cope with forcing residents to stay in their rooms, which one staff member described was “against [the resident’s] human rights”, particularly for residents with dementia. Staff mourned the loss of residents they had known for years while persevering working under these challenging conditions.

Theme 2. Lack of support

Staffing shortages and ineffective outbreak management contributed to staff feeling unsupported. Shortages were at an all-time high and workers felt there were no appropriate measures in place to ensure resident safety. Decisions to send asymptomatic workers with suspected exposure to COVID-19 home created significant challenges by leaving residents with unmet healthcare needs and safety concerns.

Staff also expressed frustration with other aspects of outbreak management, emphasizing a disconnect between the front line and governmental/institutional employees making decisions. Staff felt they were given unrealistic guidelines due to this disconnect. One staff member shared, “you come to this zoom meeting, and everybody was sitting in their offices and having sips of coffee while giving us orders. I was like, I don’t know if you know what the reality is here, why don’t you just come in and, you know, actually physically see what’s going on”. They felt they were given “impossible things to do” and compared the workplace to a “war zone”. Staff also found the rules and guidelines they were given were inconsistent and unclear, leading to confusion and frustration. Staff mentioned it is common for LTC positions to be part-time and that staff typically work at multiple sites. The single site order put in place by the Public Health Agency of Canada to control the spread of COVID-19 in LTC caused restrictions in the ability to staff to work at multiple sites and contributed to staffing shortages and increased overtime.

Later in the pandemic, the regional health authority assembled staff pools for LTC sites facing outbreaks. Staff were grateful for this program but wish it had been implemented sooner. They stressed the need for improved collaboration between entities managing outbreaks and finding ways to keep decision-makers connected to their reality. In addition, staff expressed dissatisfaction with their financial compensation given the working conditions, leading to feelings of underappreciation and a divide between the front-line and higher-level employees.

Theme 3. Providing resources

Providing resources significantly supported worker’s mental health. For example, online mental health resources helped staff feel supported by management. One facility also employed a grief specialist who held workshops on grief processing. Celebration of life events were organized to help staff cope with resident deaths.

Providing extra staff and equipment was also important in improving staff wellbeing. The LTC facilities created personal protective equipment (PPE) tables outside of each resident’s door, enhancing staff’s sense of safety and support. When staff pools from the local health authority were allocated to their site, it prevented a staffing crisis and some of the consequences described in themes 1 and 2.

Theme 4. Sense of community

Amidst the crisis caused by the pandemic, staff members demonstrated the power of community. Staff actively fostered this community spirit and it helped improve staff wellbeing. Managers played an important role in creating a positive workplace culture and ensured they were physically present to support the staff. One manager reported, “as management we were there every single day. We were delivering meals, we were taking out the garbage, we were doing all those things. So that helped with staff morale”. When referencing the fact that in some LTC facilities, management worked remotely, she emphasized the importance of management coming in person to prevent their staff from feeling abandoned, “I can’t leave. I can’t leave you there without me being there. I can’t expect you to do that without me doing that”. Managers implemented incentives to show staff they were appreciated, such as small gifts, food, and appreciative notes.

Teamwork played an important role in creating a sense of community. Staff were flexible in their roles during periods of staffing crises. For example, a music therapist would help with feeding the residents. Staff from all backgrounds worked together as a team and ensured residents were being taken care of. Staff also emphasized the importance of checking in with each other and one manager described, “staff members would come to work crying and you can’t give them a hug. So, we invented the COVID hug. You know, which is an air hug”.

One facility found success in forming consistent, smaller staff cohorts, enhancing social bonds between staff and between staff and residents. It allowed staff to “know the residents better and staff seem to be happier as well. They are almost always working with the same team”. This also helped staff feel safer because with exposure to less people each day, the risk of infection was reduced.

Additionally, residents and families played a role in this community. Social connection with residents brings staff members joy and reduces both staff and resident loneliness. One staff member, when referencing her interactions with residents described, “I take the time to have that connection with someone so asking how they’re doing … Sometimes seeing that smile [from resident] really makes me feel it’s a good day”. Family members also helped staff feel less lonely through food donations and letters to recognize the staff (“it was nice to get that to see that somebody recognized everything those staff were doing and all those long hours”).

SUPPORT Framework: Strategies to support staff working in long-term care.

The SUPPORT acronym summarizes staff recommendations to improve staff mental health and promote retention in the LTC workforce (Table 2). Since many of the problems staff faced existed prior to the pandemic, this framework may help improve the wellbeing of staff beyond the COVID-19 pandemic.

## 4. Discussion

Older adults living in LTC, and as an extension LTC staff, are often not prioritized due to ageism remaining a pervasive issue in society [30]. This was evident during the pandemic despite LTC residents having higher levels of comorbidities and being a vulnerable population [31]. There is an urgent need to strengthen the LTC workforce to address the growing needs of the aging population and improve emergency response preparedness.

This study explored the perspectives of LTC staff on factors affecting their mental health and retention in the workforce. Interviews in this study were conducted during the COVID-19 pandemic, with the goal of identifying areas of improvement during a period of heightened vulnerability. Since many of the problems faced by staff predated the pandemic, the findings of this study apply to the LTC workforce beyond pandemic settings. Building on previous research, this study engages staff to identify effective mental health support strategies and gather their recommendations to strengthen the LTC workforce. The study involves one-on-one interviews with staff working in diverse roles (nurse, managers, recreation staff, etc.) within two Canadian LTC facilities in British Columbia. Our study is unique in that it directly asked Canadian LTC staff members what they need to feel supported and improve their mental health. Most of the existing literature focuses on characterizing the experiences and severity of mental health concerns among staff and are lacking staff informed recommendations.

Four key themes emerged: (1) Depletion: staff experienced intensified challenges including staff shortages, increased workloads, isolation, lack of family visitor support, and grief, leading to task-oriented care. (2) Lack of support: workers report a disconnect with decision-makers providing unrealistic and inconsistent guidelines. (3) Providing resources: mental health support, grief workshops, PPE, and appropriate staffing levels helped improve staff wellbeing. (4) Sense of community: support from management, teamwork, connections with residents, and working in consistent small staffing cohorts fostered resilience and improved well-being. These findings informed the SUPPORT framework (Table 2), which summaries actionable recommendations for addressing staff needs.

### 4.1. SUPPORT Framework

This framework simplifies findings from the study to provide a memorable way to apply suggestions made by staff. The first part of the framework is sense of community. The staff interviewed in the study emphasized how sense of community helped support their wellbeing. Staff may be more willing to remain in the LTC sector and specific workplace if they feel they have a supportive community. Community in LTC has many elements, including frontline staff, management, caregivers, and residents. Forming a group of staff members, or cohort, that get to know each other overtime and create bonds can support this community. LTC facilities can create social events for staff such as team lunches or events outside of work for staff bonding if desired [15]. Suggestions for social events from staff can help enhance inclusivity and attendance. Due to the current lack of full-time roles in LTC, many staff members work in multiple locations and miss opportunities to grow community in one facility. However, during the pandemic the one-site order contributed to staffing shortages, since staff are normally mobile and able to fill in at other locations that have more severe shortages. Having this feedback of both wanting full-time roles for financial stability and also the need to ensure adequate staffing across facilities calls for more funding to create enough full-time roles at each site to avoid shortages even when staff are away or ill. This can also help attract workers to the LTC sector and help with the overall shortage.

Managers also play a critical role in fostering a sense of community. Proper training of management across facilities can equip managers with skills to support their staff and foster community. Strategies include promoting communication and feedback from staff. A system to collect anonymous feedback from staff members may facilitate honest communication from staff to address issues in the workplace and help staff feel heard. Managers should be encouraged to bring feedback from staff to health authorities and advocate for improvement as a key part of the community. Managers should also be encouraged to have one on one check ins with their staff members including an invitation to speak about well-being and offer support to connect to other resources if needed. Managers in our study held team meetings and checked in on staff regarding their mental health. Similar ‘huddles’ have been studied formally, such as by McGilton et al. [32]. They introduced regular huddles that addressed staff wellbeing at one LTC home in Ontario during the pandemic, resulting in staff reporting lower levels of moral distress and perceived higher levels of support.

Now that visitation restrictions are no longer in effect, family/friend caregivers of residents are back in LTC facilities. Although not a main topic discussed in our findings, supporting relationships between staff and caregivers may further create a sense of community and comfort at the LTC home. The pandemic impacted the caregiver-staff relationships, for some long-standing relationships were negatively impacted while others showed elevated support and appreciation [3]. To encourage positive relationships and contribute to community, LTC facilities should encourage open communication between staff and caregivers to ensure a mutual understanding and provide education for caregivers. Family meetings and caregiver workshops could be offered to continue to build trust and transparency. Residents also play a role in this community, which is further discussed in the context of person-centre care.

The next part of the framework is unify and educate. Frontline staff have not historically had a major voice in policy making and the pandemic has highlighted the need for change from a systems level. Satterstrom et al. also emphasize the need for the frontline voice to lead to change in healthcare [33]. The authors provide a multilevel model of how frontline voice can reach implementation that can be applied to LTC. The authors acknowledge the importance of the frontline voice, but their ideas may not be acted upon. From an individual level, frontline workers need to feel empowered and motivated to voice their concerns. This is where LTC managers can use the supportive community environment to create a space where staff feel encouraged to voice their ideas and concerns. Both the team and organizational level also have a role in creating change from frontline voices. Power dynamics, limited resources, organizational norms, and lack of an ‘influential advocate’, individuals who have power to champion frontline ideas and push for action, are obstacles that need to be addressed to ensure ideas from frontline workers are implemented.

Positive staff morale is the next pillar in the framework. Positive morale can stem from someone’s inherent character and dedication to their profession. For example, some staff report a sense of duty and commitment to the care of residents and find value and fulfilment in their work [34,35]. However, positivity can be fostered at multiple levels in the system. In our study, positive morale in LTC homes was supported by acts and tokens of gratitude, providing leadership opportunities to frontline workers, and ensuring clear communication. Acts of gratitude was similarly cited in other work in contributing to positivity [15], although in context of extra COVID-19 pay, incentives and healthcare benefits such as massages. Optimizing external factors can help support positive morale among LTC staff.

Person-centered care in LTC supports relationships between residents and staff and promotes satisfaction and value in LTC work. It also can ensure quality and meaningful care, which helps improve staff wellbeing. This is supported by other work which emphasizes the relationship between staff and residents and how this can enhance staff resilience [5,15]. Proper respect and recognition of LTC workers can come in the form of verbal appreciation from management, inviting staff in conversations related to decision-making, and financial compensation [15]. This study as well as existing literature are in support of providing mental health resources for staff including counselling [36] and mental health days [37]. The importance of teamwork in supporting staff mental health is also cited elsewhere [15,16,38].

### 4.2. Comparison with the Literature

These findings align with previous research documenting emotional and mental distress among LTC workers, related to staffing shortages [9,10,21], increased demands and workload [4,12,20], feelings of fear towards the virus [4,13,21], grief over loss of residents [12,21], and missing support from family visitors [22,39]. Moral distress discussed in other studies [22,23] aligns with similar experiences in this cohort relating to guilt and grief. Our study provides more evidence of these factors and highlights how universal these unfortunate experiences are for LTC workers across Canada. Commonly used terms to describe experiences such as ‘distress’, ‘burnout’, or even ‘depleted’ as in this study are likely undervaluing the chronic suffering LTC workers faced and increased rates of psychiatric illness (e.g., post-traumatic stress disorder or acute stress reaction) that can be sensed in the staff quotations in our findings. Although psychiatric illness was not assessed directly, other studies show increased rates of psychiatric illness among LTC staff related to the pandemic [19].

Findings that were more emphasized in our cohort include task-oriented care, social isolation, and disconnect with governing institutions. Staff describe a link between distress and task-oriented care that resulted from increased workloads. This type of care focuses on completing tasks, compared to person-oriented care that focuses on the resident as an individual and involves more social connection and relationship building. This demonstrates the need for staff to have social connection with residents to enjoy and find satisfaction in their work. A separate but related finding was the sacrifice staff made by socially isolating themselves outside of work to keep residents safe. This was mentioned briefly by Savage et al., in which managers asked their staff to limit social contact outside of the home [20]. Our cohort emphasized the impact of this isolation on their mental health.

Other work focuses on the impact of rapidly changing guidelines [4], while the staff in our study emphasized the unrealistic nature of guidelines due to disconnect with policy makers and health authorities. Interestingly, care aids from another site in British Columbia emphasized a lack of communication and in-person support from managers specifically [21]. In the study by Fisher et al. (2021), a recommendation from staff to care home leadership which could improve their well-being is to have more transparent communication [13]. In our study, workers called attention to the lack of communication from health authorities instead of managers. Managers showed up in person daily and prioritized keeping their team updated. This emphasizes inter-facility differences and the systematic change that needs to occur. Orhierhor et al. similarly found LTC staff in British Columbia were frustrated due to poor communication and coordination between health authorities and lack of involvement in the frontline in pandemic planning [40]. Efforts to educate policymakers on the realities of front-line experiences may help facilitate a deeper understanding and improve future guidelines and expectations. Clear, concise, and consistent guidelines from a single provincial organization that provides tips and resources for guideline implementation can help streamline coordination and communication [41]. To help facilitate communication within the care home, managers may implement daily huddles [32], repetition of key information, and ensure all managers are sending the same message [42].

For the LTC facilities in our study, online counselling services helped support staff mental health. Unfortunately, this service is not available at many LTC facilities. LTC facilities should reduce obstacles in accessing this resource, such as stigma or financial barriers. Reynolds et al. report only 13% of participants in their study accessed mental health supports and most respondents reported barriers in seeking help [4]. Anonymous counselling should be offered (either free or low-cost service) for LTC workers given the stressors related to their work. Newly graduated healthcare workers may particularly benefit from LTC workplace mental health supports and interventions [41].

The LTC facilities in our study were unique in organizing grief workshops and celebration of life events for their staff, which helped them cope during pandemic. Beyond the pandemic, staff must cope with the death of residents they built relationships with, provide end of life care, and provide care to residents with challenging diagnoses, such as dementia. Mental health services can help support a healthy workforce moving forward. Daily huddles as discussed previously can help staff have a platform for communication. Implementing frequent debriefs, particularly after challenging events, can give staff an opportunity to discuss processes and outcomes, while normalizing taking time to ask staff how they feel and how they can be supported moving forward.

Perceived adequacy of workplace infection control procedures and PPE has been associated with mental health symptoms among healthcare workers in Canada [43]. Our study supports the role of supplying appropriate safety equipment in supporting worker’s mental health. For understaffing concerns, staff pools from the local health authority were allocated to LTC sites in later stages of the pandemic in this study, which helped significantly with the staffing crisis. This strategy should be included in pandemic response plans. Pre-pandemic staffing levels were a concern; thus, changes are needed to improve wages, working conditions, and level of training to attract and retain LTC staff.

One of the key learning points from our findings is the role of community in supporting wellbeing. Management was actively involved during the pandemic and their physical presence played an important role in fostering community. This is different to experiences of staff from other facilities, in which they felt they rarely saw or spoke to managers directly [21]. Our findings more closely align with the experiences of registered practical nurses (RPNs) working in LTC facilities in Ontario [16]. Some of the RPNs report management was more present during the pandemic than ever, and their leadership played a key role in helping them feel supported. They describe two different experiences across sites, with some RPNs reporting their team did not band together, while other’s report their team was like family. The latter echo’s the views of a strong sense of community described in our study. A physically present, stable, caring, and supportive manager played a crucial role in supporting the LTC workers in our study. Efforts to create a positive workplace culture, where staff members help each other with tasks and work as a team to keep residents safe and cared for can help foster this sense of community. This teamwork was emphasized by both management and frontline staff in our study, since we had the opportunity to explore and compare perspectives.

Small efforts, such as managers writing appreciation notes for staff to recognize their hard work or checking in with them and offering an air “COVID hug” also supports a sense of community where each team member is seen as important. A care home in Ontario also reported the benefits of recognition in which managers ensured all staff were respected and everyone was seen as integral team members [42]. Providing appropriate training for management can help foster strong leadership skills and promote this sense of community across different LTC sites, thereby strengthening the workforce.

Another key finding that supported staff well-being and retention in our study was forming consistent staff cohorts to promote teamwork, social connection, and sense of safety. There may be a shift to smaller, homelike LTC homes in British Columbia due to the benefit of consistent staffing to both residents and staff [44]. This may also make LTC jobs more desirable because smaller homes may offer full-time jobs, which is preferable over many LTC staff’s current situation of working part-time hours at multiple locations. In addition to fostering relationships with other staff, staff can build stronger relationships with a smaller group of residents.

### 4.3. Contribution to the Literature

Our study provides evidence to support existing literature to help advocate for improving LTC. Our findings also describe an emphasis on factors contributing to poor mental health outcomes that are not as prominent in the literature, such as task-oriented care, social isolation, and disconnect with governing bodies. Our findings advocate for involving frontline workers in policy making and other decisions affecting LTC.

We wanted to give a voice to LTC staff and ask them directly what they need going forward to support their mental health and want to remain working in the field. Staff stories and suggestions directly informed the SUPPORT framework. This framework covers both what worked for LTC staff who were interviewed in this study, and what resources they wish they had. Recommendations are made both at a facility, health authority region, and provincial government level.

Our cohort emphasized the need for removing barriers to mental health services (e.g., counselling, grief workshops, celebration of life events), strong leadership that help them feel valued and heard, a positive workplace environment, and consistent staff cohorts. Their experiences can help improve emergency response plans, such as preparing staff pools to allocate to facilities in need. These supports that are needed for a healthy LTC workforce are only possible with appropriate funding and resources that require a shift in the public and government perspective on resource allocation.

This work advocates for recommendations outlined in SUPPORT to strengthen the workforce. The COVID-19 pandemic forced the public and government to learn about problems that have been worsening for years. As the public and government’s attention shifts away from the pandemic and onto new concerns, we stress the importance of learning from our frontline workers and making real change. We hope by continuing to bring attention to this issue with our study and emphasizing the need for improvement, we can play a role in informing and promoting this change.

### 4.4. Limitations

Since participants were recruited only from two LTC facilities in Vancouver, British Columbia, the results do not reflect experiences of LTC staff across other regions, provinces, or countries. This is a possible lack of transferability of the study findings and recommendations. Recommendations from this study should be assessed in context with other literature and discussed as a team for particular LTC homes. Interviews were only conducted in English and, unfortunately, we were not able to include non-English speaking participants, although we were not aware of any participants who this applied to. Including more participants, particularly those with different racial and ethnic backgrounds, gender, and age may have provided information from a broader range of experiences in LTC.

### 4.5. Implications for Future Research

Next steps for research should focus on the second phase of the CAR process, which involves collaboration with stakeholders to implement these findings and recommendations from the frontline in LTC. Policy making should evaluate this study alongside other literature to create improvements for LTC employees with mental health on the forefront of planning. Future pandemic response plans would also benefit from this feedback collected from staff. Larger sample sizes should be used to test the recommended strategies and ensure feasibility in LTC.

## 5. Conclusions

This qualitative study highlights the urgent need to improve resources and mental health support for the LTC workforce. Attention towards LTC is critical to promote retention of staff members, who play a critical role in caring for an aging and vulnerable population. Pre-pandemic problems were exacerbated by COVID-19 outbreaks and emphasized the factors contributing to poor mental health outcomes among staff. The SUPPORT framework was developed based on the perspectives of front-line staff to offer practical tips in supporting the wellbeing and retention of staff. Ensuring staff feel valued and supported is crucial, not just for future pandemic responses but for long-term resilience in the LTC workforce.

## Figures and Tables

**Table 1 healthcare-13-00040-t001:** Descriptive characteristics of participants.

Participant Characteristics	%
Age (years)	
Younger than 35	20
36–50	60
Older than 50	20
Gender	
Male	20
Female	80
Other	0
Ethnicity	
Caucasian	40
South Asian	60

**Table 2 healthcare-13-00040-t002:** SUPPORT is a staff-informed framework that summarizes practical strategies to enhance mental health and improve retention among LTC staff.

	Description	Specific Tools and Recommendations
Sense of community	When staff feel like they are a valued part of a community, it promotes connection and reduces feelings of isolation.	Foster a sense of community with regular check-ins or huddles. Ask how staff are doing, be attentive and kind when listening, and take action to respond to issues. Create more full-time positions at a single site and form consistent staff cohorts that work with the same group of residents each day. This can promote teamwork and social connection amongst staff and between staff and residents.
Unify and educate	Staff are impacted by decisions made for long-term care facilities (i.e., by health authorities, policymakers).	Focus on educating these entities about the realities in long-term care (e.g., including in-person visits and status updates) to inform policy, resource allocation, and public health measures. Bring various entities together to encourage communication, collaboration, and understanding. Offer a “seat at the table” for frontline representatives to help with decision-making.
Positive staff morale	Positive morale in the workplace boosts pride, productivity, and job satisfaction.	Build positive morale by offering tokens of gratitude such as thank you cards, goodie bags, or food. Support staff-led initiatives to foster leadership and prioritize clear communication to staff.
Person-centered care	Staff members feel fulfilled when they can connect meaningfully with residents.	Educate staff about person-centered care and equip them with tools to engage effectively with residents. Ensure staffing levels are adequate to allow time for connecting with residents.
Offer respect and recognition	Feeling respected and heard is essential.	Listen and respond to staff needs. Ensure staff participate in decision making. Show appreciation for staff by offering recognition and explain the reason behind your appreciation. Prioritize proper financial compensation for workers.
Resources for mental health support	Caring for mental health is necessary, including for those who provide care to others.	Offer culturally appropriate mental health services and mental health days. Provide staff with tools to support their wellbeing daily, such as deep breathing or meditation exercises.
Teamwork	Teamwork fosters supportive and lasting relationships.	Create staff cohorts to allow staff to work with and get to know the same team and group of residents over time. Encourage staff to help other workers when needed and appropriate.

## Data Availability

The datasets generated and/or analysed during the current study are not publicly available due to a reason of protecting participants’ privacy but are available from the corresponding author on reasonable request.

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
