# Peer review of "Behind the Frontlines: Insights for Supporting Mental Health and Staff Retention in the Long-Term Care Workforce"

_healthcare, 2024, doi:10.3390/healthcare13010040_

Round 1
Reviewer 1 Report
Comments and Suggestions for Authors
Thank you for providing me the opportunity to review your work, which no doubt was not an easy task during that period. Although a few years ago, the pandemic has left lasting effects on the healthcare profession worldwide and this work is still a relevant piece. The initial findings revealed very perceptions of staff and ways of working ie online in comparison to the reality of hard work in the care homes. The theme a lack of support still prevails in healthcare services worldwide and this theme is of most importance. The academic quality of this work is of a good standard and the English grammar is met. The narration structure and methodology is clear. An array of available studies explored the impact of the pandemic on the healthcare environment. There were notable systemic finds surrounding morale distress and impact on LTC staff.
‘The research team consisted of seven women and one man’ perhaps female and male? Only a suggestion to amend.
Author Response
Comment 1: The research team consisted of seven women and one man’ perhaps female and male?
The wording of women/man has been changed to female and male (page 5 line 245). Thank you for the suggestion and for your review.
Reviewer 2 Report
Comments and Suggestions for Authors
The authors present a study aimed at exploring the experience of Covid-19 and the opportunities to promote workplace health among long-term care workers. It is a cross-sectional qualitative study involving 16 Canadian long-term care workers. The findings are interesting as they delve into the critical situations faced by this category of workers during the pandemic and also highlight some possible strategies to improve their overall well-being. The manuscript is generally well structured and well written. Nevertheless, some minor revisions could improve its quality and make it suitable for publication in this journal.
Below are my comments on the sections of the manuscript that need improvement.
---------------------------------------------------------------------------------------
Abstract
- In line 18, the number of participants should be added (i.e. sixteen staff members), and the reference to the participants in line 20 can be removed.
Introduction
- As one of the most important resources examined by the authors, more space should be devoted to resilience in the introductory section.
- It would be interesting to provide some specific information on how work in long-term care has changed during the pandemic.
Materials and Methods
- As for the analysis conducted, did the authors use some a priori themes derived from the literature (or a theoretical framework) or did they use a grounded approach?
- It would be helpful to add some of the questions that were asked in the interview.
- Lines 128-129: What does it mean that the study “includes experiences from both outbreak and non-outbreak settings”? Was the study not conducted during the pandemic?
- Was the sensitive data in the transcriptions anonymized as they were also read by external members (as per the paragraph about the “research team”)?
Author Response
Comment 1: In line 18, the number of participants should be added (i.e. sixteen staff members), and the reference to the participants in line 20 can be removed.
This has been changed on line 18 and 20, thank you for the suggestion.
Comment 2: As one of the most important resources examined by the authors, more space should be devoted to resilience in the introductory section.
This is a wonderful idea. On page 2, line 91-112, we have dedicated more space to discuss resiliency in the long term care workforce.
Comment 3: It would be interesting to provide some specific information on how work in long-term care has changed during the pandemic.
We have added more specific information on how long-term care has changed during the pandemic on page 1 line 39-56. Thank you for the suggestion.
Comment 4: As for the analysis conducted, did the authors use some a priori themes derived from the literature (or a theoretical framework) or did they use a grounded approach?
We used a grounded approach. This was made clear on page 5 line 272-273.
Comment 5: It would be helpful to add some of the questions that were asked in the interview.
Please see Appendix A for the guiding questions on page 15 line 879.
Comment 6: Lines 128-129: What does it mean that the study “includes experiences from both outbreak and non-outbreak settings”? Was the study not conducted during the pandemic?
Yes the study was conducted during the pandemic. At certain times during the pandemic there was active outbreaks (i.e., multiple cases of COVID-19 positive residents) and times when there were no positive cases, i.e., non-outbreak setting. We have removed this comment on lines 128-129 to avoid confusion.
Comment 7: Was the sensitive data in the transcriptions anonymized as they were also read by external members (as per the paragraph about the “research team”)?
Yes, the data was anonymized. We clarified this on page 5 line 260. Thank you for your suggestions and feedback on helping improve this manuscript.
Reviewer 3 Report
Comments and Suggestions for Authors
The article addresses the important topic of the staffing crisis and mental health issues of long-term care (LTC) staff in Canada, especially in the context of the COVID-19 pandemic. These issues are timely and critical to the quality of care for the elderly and the sustainability of the health care sector.
Originality and presentation:
Although the topic is relevant, the article does not explicitly introduce innovative conclusions or significantly expand existing theoretical knowledge. The introduction of the SUPPORT framework as a proposal for intervention is an interesting element, but its presentation remains underdeveloped. Also missing is a broader theoretical analysis that sets the findings in the context of the existing literature.
Assessment of content quality:
The article describes the findings in general terms and not in sufficient detail, which weakens the strength of the argument. The identification of four themes (exhaustion, lack of support, provision of resources and sense of community) is valuable, but the analysis of these aspects seems superficial. The thematic analysis, although based on interviews with staff members, was not sufficiently substantiated theoretically or practically. There is a lack of information on how the themes identified fit into the literature and what empirical evidence supports the SUPPORT framework presented.
The conclusions are logical, but lack depth of analysis and clear reference to the literature. It remains unclear how the SUPPORT framework can be implemented in practice and how it translates into existing policies or theories.
Evaluation of methodological aspects:
The article uses appropriate terminology, but lacks sufficiently detailed methodological definitions. The Collaborative Action Research (CAR) methodology and its various steps in the context of this study are not explained precisely. The selection of methods, such as individual interviews and thematic analysis, is reasonable, but no details are provided regarding the selection of the research sample. No explanation is given as to why these two long-term care homes were chosen and what the selection criteria were. Information about the data analysis tools, such as the software used, is also missing.
Citation and use of sources:
The literature used in the article is correctly selected, but poor, and does not include recent items in the field of mental health research on LTC staff. In the introduction, the contributions to the literature are incorrectly presented. The authors write: “To contribute to this literature, this paper will present one-on-one interviews with a diverse range of staff....” - conducting a study is not in itself a scientific contribution if it is not accompanied by a solid theoretical analysis and reference to existing work.
General comments
The article raises important issues related to the mental health of LTC staff, but leaves a lot unsaid in terms of theoretical justification, methodology and analysis of results. The biggest shortcomings include:
1. Lack of a clearly defined research problem and research gap.
2. Inadequate description of Collaborative Action Research methodology and data analysis procedures.
3. Superficial analysis of the results that does not adequately address the literature.
Despite these limitations, the article is an important step toward integrating the mental health of LTC staff into policies and practices. However, its potential impact on theory and practice is limited by the lack of innovation and detail in the analysis. Further work should focus on deepening the theoretical analysis and describing the methodology in more detail.
Author Response
Comment 1: Although the topic is relevant, the article does not explicitly introduce innovative conclusions or significantly expand existing theoretical knowledge. The introduction of the SUPPORT framework as a proposal for intervention is an interesting element, but its presentation remains underdeveloped. Also missing is a broader theoretical analysis that sets the findings in the context of the existing literature.
Thank you for this comment. We agree the SUPPORT framework is a new element we can bring to the literature. We find value in an acronym especially in clinical practice as a memory tool and simple break down of a large amount of findings/recommendations. We have not yet seen an acronym used in articles describing recommendations to support LTC staff wellbeing. We discussed the SUPPORT framework in depth with our research team but agree we can bring more of what what discussed in the analysis to the paper. To address this, we included further discussion of the SUPPORT framework on page 10 line 558. We related each element to existing literature.
We appreciate that the presentation of the study could benefit from including more literature in the discussion. We initially intentionally kept the majority of the referenced literature limited to Canadian studies as policies and COVID-19 pandemic responses varied across the world and wanted to focus our lens on Canada as one example. However we understand there are many shared experiences in LTC across the world and we will have an international audience. Thus, we have integrated further international literature into the discussion. Line 65 on page 2 in the introduction provides more background and international studies are integrated in the discussion on page 11 lines 638-650. Thank you for the suggestion.
Comment 2: The article describes the findings in general terms and not in sufficient detail, which weakens the strength of the argument. The identification of four themes (exhaustion, lack of support, provision of resources and sense of community) is valuable, but the analysis of these aspects seems superficial. The thematic analysis, although based on interviews with staff members, was not sufficiently substantiated theoretically or practically. There is a lack of information on how the themes identified fit into the literature and what empirical evidence supports the SUPPORT framework presented. The conclusions are logical, but lack depth of analysis and clear reference to the literature. It remains unclear how the SUPPORT framework can be implemented in practice and how it translates into existing policies or theories.
We appreciate your comment on the analysis of the themes. We would like to bring attention to Appendix B on page 15 line 886 which summarizes the identification of themes based on the data for more clarity on how these themes were developed. Further discussion of the SUPPORT network on page 10 line 558 provides more practical suggestions for implementing these findings. We also provide further detail about the literature that aligns with the SUPPORT framework in this section. We refrained from repeating similar arguments and concepts that are outlined later in the discussion, since the SUPPORT framework is based on the themes described.
Comment 3: The article uses appropriate terminology, but lacks sufficiently detailed methodological definitions. The Collaborative Action Research (CAR) methodology and its various steps in the context of this study are not explained precisely. The selection of methods, such as individual interviews and thematic analysis, is reasonable, but no details are provided regarding the selection of the research sample. No explanation is given as to why these two long-term care homes were chosen and what the selection criteria were. Information about the data analysis tools, such as the software used, is also missing.
We have added more information about the CAR process on page 4 lines 190-195. We emphasized what the 'collaboration' part of the process entails and how our study team members played an important role in this step. We also highlighted why the two LTC homes were invited to participate in the study on page 5 line 231-234. There was no specific software platform used. We appreciate these comments to help clarify our methodology.
Comment 4: The literature used in the article is correctly selected, but poor, and does not include recent items in the field of mental health research on LTC staff. In the introduction, the contributions to the literature are incorrectly presented. The authors write: “To contribute to this literature, this paper will present one-on-one interviews with a diverse range of staff....” - conducting a study is not in itself a scientific contribution if it is not accompanied by a solid theoretical analysis and reference to existing work.
We have added in more international literature in the introduction (starting on line 65 on page 2) and discussion (page 11 lines 638-650). We are happy to add additional literature, particularly if there are certain aspects of the literature you believe we should address. We agree with your comment and have rephrased the presentation of the article. Thank you kindly for your suggestions to improve our manuscript.